# Orbital Adipose Tissue: The Optimal Control for Back-Table Fluorescence Imaging of Orbital Tumors

**DOI:** 10.3390/bioengineering11090922

**Published:** 2024-09-14

**Authors:** Lan Yao, Wenhua Zhang, Xuedong Wang, Lishuang Guo, Wenlu Liu, Yueyue Li, Rui Ma, Yan Hei, Xinji Yang, Zeyu Zhang, Wei Wu

**Affiliations:** 1Senior Department of Ophthalmology, 3rd Medical Center of Chinese PLA General Hospital, Beijing 100143, China; yl1845503302@163.com (L.Y.); nku1710999@163.com (W.L.); liyueyue_7909@163.com (Y.L.); marui860910@163.com (R.M.); heiyan842@163.com (Y.H.); 2Research Center for Tissue Repair and Regeneration Affiliated to the Medical Innovation Research Division, 4th Medical Center of Chinese PLA General Hospital, Beijing 100048, China; wenhuazhang304@outlook.com; 3Beijing Tsinghua Changgung Hospital, School of Clinical Medicine, Tsinghua University, Beijing 102218, China; wangxuedong301@163.com; 4Key Laboratory of Big Data-Based Precision Medicine, Ministry of Industry and Information Technology, Beijing 100191, China; gls821@buaa.edu.cn; 5Beijing Advanced Innovation Center for Big Data-Based Precision Medicine, School of Engineering Medicine, Beihang University, Beijing 100191, China

**Keywords:** orbital tumor, fluorescence imaging, NIR-II, back-table imaging, image-guided surgery

## Abstract

Control tissue is essential for ensuring the precision of semiquantitative analysis in back-table fluorescence imaging. However, there remains a lack of agreement on the appropriate selection of control tissues. To evaluate the back-table fluorescence imaging performance of different normal tissues and identify the optimal normal tissue, a cohort of 39 patients with orbital tumors were enrolled in the study. Prior to surgery, these patients received indocyanine green (ICG) and following resection, 43 normal control tissues (34 adipose tissues, 3 skin tissues, 3 periosteal tissues, and 3 muscle tissues) were examined using back-table fluorescence imaging. The skin tissue demonstrated significantly elevated fluorescence intensity in comparison to the diseased tissue, whereas the muscle tissue exhibited a broad range and standard deviation of fluorescence signal intensity. Conversely, the adipose and periosteum displayed weak fluorescence signals with a relatively consistent distribution. Additionally, no significant correlations were found between the signal-to-background ratio (SBR) of adipose tissue and patients’ ages, genders, weights, disease duration, tumor origins, dosing of administration of ICG infusion, and the time interval between ICG infusion and surgery. However, a positive correlation was observed between the SBR of adipose tissue and its size, with larger adipose tissues (>1 cm) showing an average SBR 27% higher than smaller adipose tissues (≤1 cm). In conclusion, the findings of this study demonstrated that adipose tissue consistently exhibited homogeneous hypofluorescence during back-table fluorescence imaging, regardless of patient clinical variables or imaging parameters. The size of the adipose tissue was identified as the primary factor influencing its fluorescence imaging characteristics, supporting its utility as an ideal control tissue for back-table fluorescence imaging.

## 1. Introduction

Precise resection is crucial for the treatment of orbital tumors, given the complex anatomy of surrounding nerves, muscles, and tissues. However, distinguishing between tumor and healthy tissue through visual and tactile inspection alone poses challenges. The real-time intraoperative identification of tumors is essential to improve the precision of surgery [1,2,3]. In recent years, the widespread adoption of fluorescence imaging technology in tumor surgeries has proven beneficial in aiding surgical decisions and improving patient outcomes. Specifically, fluorescence imaging in the second near-infrared window (NIR-II, 900~1880 nm) based on indocyanine green (ICG) has been shown to enhance the sensitivity and precision of intraoperative tumor detection, presenting notable advancements in clinical practice [4]. Following intravenous administration, ICG infiltrates and gathers in tumor lesions, emitting distinctive NIR-II fluorescence with heightened contrast due to variances in metabolic activity between tumor and normal cells.

In clinical settings, the utilization of the back-table fluorescence imaging technique is common for conducting quantitative analyses on excised tissue, resulting in fluorescence imaging outcomes that closely correlate with the precision of pathology. Back-table fluorescence imaging involves capturing images of excised tissues within standardized close-field imaging conditions, followed by a quantitative analysis and objective assessment of the lesion’s nature [5,6,7,8,9,10]. This approach enables a highly sensitive and semiquantitative evaluation of resection margins, within 1 min after specimen excision in the surgical theater [11,12], leading to its widespread acknowledgement and adoption in intraoperative tumor identification [13,14,15,16].

In order to enhance the precision of the semiquantitative analysis in back-table fluorescence imaging, researchers commonly select normal tissue as the control to mitigate the impact of tissue autofluorescence, photon scattering, photon absorption, and fluorescence imaging conditions. Previous studies have utilized muscle tissue [17], fibro-adipose [18], adipose tissue [17], and surrounding normal tissue [8,12,19,20] as control samples. However, the optimal control tissue for back-table fluorescence imaging has not been conclusively determined, primarily because of a lack of research on the fluorescence imaging performance of various normal tissues [21,22,23]. This limitation hinders the precision and scalability of back-table NIR-II fluorescence imaging.

Optimal control tissue for back-table fluorescence imaging should possess the following features: (1) significantly lower fluorescence intensity compared to diseased lesions; (2) uniform and stable distribution of fluorescence signal; (3) easy accessibility; and (4) intraoperative resection without causing additional harm to patients. In this study, we performed NIR-II fluorescence imaging-guided orbital tumor surgery, and compared the fluorescence imaging characteristics of various control tissues (adipose, periosteum, muscle, and skin), providing a basis for selecting ideal control tissues for fluorescence imaging.

## 2. Methods and Materials

### 2.1. Ethics

This research was conducted in accordance with the principles outlined in the Declaration of Helsinki, as approved by the ethics committee of the 3rd Medical Center of the Chinese PLA General Hospital (2020-004), and registered with the Chinese Clinical Trial Registry (ChiCTR2000039908). Prior to data collection, all participants were provided with detailed information regarding the study’s objectives, experimental procedures, potential adverse events, and other relevant details, and provided written consent.

Thirty-nine participants were enrolled from the Senior Department of Ophthalmology at the 3rd Medical Center of the Chinese PLA General Hospital between January 2022 and March 2023. Inclusion criteria stipulated that patients must be 18 years or older, clinically diagnosed with orbital solid tumors by two experienced orbital physicians, and scheduled for standard-of-care surgery. Exclusion criteria encompassed individuals under the age of 18, those with allergies to iodine or shellfish, and those deemed unfit for surgical intervention.

### 2.2. Back-Table Fluorescence Image Analysis

The home-built back-table fluorescence imaging system was composed of a liquid-cooling InGaAs charge-coupled camera (CCD, Cheetah 640, Xenics, Leuven, Belgium), integrated with a large depth-of-field lens (focal length 50 mm and maximum iris 1.8) and an 1100 nm longpass optical filter (FELH1100, Thorlabs, Newton, NJ, USA). A 792 nm laser generator (output power 2.0 W, Changchun Leirui Optoelectronic Technology, Changchun, China) was used to excite the fluorophores. The CCD camera and laser generator were fixed on the top of a black box, which occluded ambient light.

Prior to surgery, patients were administered an intravenous infusion of about 0.7 mg kg (1 body weight of ICG (Dandong Yichuang Pharmaceutical, Dandong, China)) 2 h before undergoing a resection of the orbital tumor. Following the removal of the tumor, adjacent normal tissues such as adipose, periosteum, muscle, and skin were excised approximately 5 mm from the surgical margin. The excised specimens were then subjected to imaging using the back-table NIR-II fluorescence imaging system. Regions of interest (ROIs) were delineated on the excised samples to calculate their mean fluorescence intensity (MFI). The SBR was quantitatively calculated by comparing the MFI of the excised tissue with that of black paper placed underneath the tissue, which served as the background.

### 2.3. Histopathology

A histopathology assessment served as the gold standard for the diagnosis of the resected samples. The samples were fixed in formalin, embedded in paraffin, and sequentially sliced at a thickness of 5 μm. Subsequently, the sections were subjected to staining with hematoxylin and eosin (H&E) following established protocols.

## 3. Results

### 3.1. Back-Table Fluorescence Imaging Characteristics of Control Tissues

This study included 39 patients diagnosed with orbital tumors, with an average age of 52.9 years (range 18–80 years) and 20 male patients (51.28%). The average weight of the patients was 67.2 kg, with a range of 47–92 kg (Appendix A). Normal tissue adjacent to the tumor was chosen as a control for objective and a quantitative back-table fluorescence imaging analysis of excised orbital tumor samples. A total of 34 adipose tissues, 3 skin tissues, 3 periosteal tissues, and 3 muscle tissues were obtained for analysis, all of which were confirmed to be free of tumor cells through postoperative pathological examination.

Firstly, we analyzed the fluorescence intensity of these control tissues. The patient-matched skin tissue displayed a higher MFI than the tumor tissue (Figure 1A), making it unsuitable as a normal control. Conversely, adipose tissue, muscle, and periosteum exhibited lower fluorescence levels compared to the tumor tissues (Figure 1B–D). A subsequent examination of the fluorescence signal intensity distribution in adipose tissue, periosteum, and muscle revealed that the muscle tissue exhibited the widest range and standard deviation of fluorescence signal intensity (Figure 1E,F). This observation suggested an uneven distribution of fluorescence signal in muscle tissue, rendering it unsuitable as a standard control for back-table fluorescence imaging. In contrast, both periosteum and adipose exhibited hypofluorescence with a more uniform signal distribution, exhibiting ideal fluorescence signal characteristics of the control tissue. Given the wider distribution of adipose tissue in the orbit, and its resection is innocuous, we posit that adipose tissue is a more suitable control tissue for back-table fluorescence imaging than periosteum. Consequently, in the following study, we used orbital adipose as the research object.

### 3.2. Effect of Basic Clinical Characteristics of Patients on Fluorescence Imaging Results of Orbital Adipose

In order to investigate the impact of patients’ general characteristics and their disease status on the SBR of back-table fluorescence imaging of orbital adipose, we compared the fluorescence imaging outcomes among patients of varying ages, genders, weights, disease durations, and tumor origins. The findings indicated that these clinical variables did not result in substantial alterations in the SBR of back-table fluorescence imaging of orbital adipose (Table 1), indicating that the fluorescence imaging signal of orbital adipose did not depend on the patients’ general clinical characteristics.

### 3.3. Effect of ICG Administration Method on Fluorescence Imaging Results of Orbital Adipose

We further identified the potential effects of imaging conditions on the fluorescence imaging results of orbital adipose, focusing on the dosing of the administration of ICG infusion and the time interval between ICG infusion and surgery. Our initial plan was to administer ICG at a dosage of 0.7 mg/kg body weight before the experiment. However, due to constraints related to the clinical use of ICG, the actual dosage varied among patients (ranging from 0.56 to 0.72 mg/kg). We evaluated the total amount of ICG administered to each patient and analyzed its association with SBRs. No significant correlations were found between the dosing strategy (*p* = 0.5519 for fixed dosing and *p* = 0.6531 for weight-based dosing) (Figure 2A,B). Additionally, the patients in our study underwent surgery at a mean time of 3.6 h (range 2–10 h) following ICG injection. There was also no significant correlation between the SBRs and the time from ICG infusion to surgery (*p* = 0.5284) (Figure 2C). Thus, it was concluded that the infusion dose and time from ICG infusion to surgery did not impact the fluorescence imaging results of orbital adipose in our study.

### 3.4. Effect of Adipose Tissue Size on Its SBR

The impact of adipose tissue size on the SBR of back-table fluorescence imaging was further investigated in this study. An analysis using Spearman’s correlation revealed a statistically significant relationship between adipose tissue size and the SBR (ρ = 0.53 and *p* = 0.0014). Subsequent grouping of the 34 adipose tissues demonstrated that larger adipose tissues (>1 cm) exhibited an average SBR that was 27% higher than smaller adipose tissues (≤1 cm) (Figure 3A,B). These findings suggested that adipose tissue size played a role in influencing the results of back-table fluorescence imaging.

## 4. Discussion

Fluorescent molecular imaging has become an important means for optimizing surgical decisions in cancer treatment. However, during intraoperative imaging procedures, noise signals may arise from the environment, imaged tissues, or imaging equipment, which can affect the accuracy of fluorescence imaging. Over the past decade, researchers have continuously improved fluorescent molecular imaging techniques, including the design of high-performance imaging molecules [24] and other non-invasive imaging technologies [25,26]. In this study, we utilized NIR-II back-table intraoperative imaging technology to better quantitatively analyze the ex vivo fluorescence imaging results of tumor tissues.

The selection of appropriate control tissue is crucial for the quantitative analysis and imaging accuracy of back-table fluorescence imaging [16]. The standardized selection of control tissue can facilitate the clinical implementation of this technology. Nevertheless, there is currently a lack of consensus on the recommended selection of suitable control tissues and limited research on the fluorescence imaging characteristics of normal control tissues. Hence, in the context of orbital tumor surgery, we conducted a study to gather and analyze the fluorescence properties of four typical control tissue types. Our findings indicated that adipose tissue served as an ideal control tissue for fluorescence imaging, potentially enhancing the navigation of head and neck tumors and their related lesions.

In the selection of control tissues, we recommend considering three key characteristics: intuition, consistency, and accessibility. The human eye is capable of discerning high fluorescence signals from low fluorescence signal areas with greater sensitivity [27]. Therefore, it is posited that an ideal control tissue should exhibit a significantly lower fluorescence signal intensity compared to the target tissue. Additionally, the variability in fluorescence signal distribution among tissues may lead to sampling errors [28]. Consequently, the ideal control tissue should demonstrate a consistent fluorescence signal distribution to minimize such errors and ensure the reliability of fluorescence imaging outcomes. Finally, it is imperative that the control tissue be readily available without compromising patient well-being or surgical protocols.

Following a thorough assessment based on these criteria, skin tissue was deemed unsuitable as an ideal control tissue due to its hyperfluorescence. Consistently, the ICG-based fluorescence imaging of head and neck cancers has also shown that the fluorescence signal intensity of skin tissue is comparable to or greater than that of the diseased tissues [29,30]. Moreover, fluorescence imaging using panitumumab-IRDye800CW in head and neck cancers has also revealed that skin tissue displays comparable fluorescence intensity to diseased tissue [31]. The heightened fluorescence signal in skin tissue may be attributed to glycosylation products or other fluorescent substances present on the skin surface [32]. Conversely, adipose tissue exhibits significantly lower fluorescence intensity compared to diseased tissue, with a uniform signal distribution and stable fluorescence imaging results that are unaffected by patient clinical characteristics or imaging conditions. Consistently, Eben L. Rosenthal and colleagues have also shown that the fluorescence signal intensity of adipose tissue is lower compared to diseased tissue [33]. We demonstrated variations in fluorescence SBRs between large and small adipose tissues, potentially attributable to the differences in ICG contents. This observation aligns with previous research findings that establish a robust association between the tissue weight and fluorescence intensity [34]. Therefore, maintaining a consistent size of excised adipose tissue is crucial for back-table fluorescence imaging. This study contributes to enhancing the precision of NIR-II back-table imaging for tumor identification and facilitates its clinical application.

Limitations: (1) The quantity of muscle and periosteum is considerably lower compared to adipose tissue because muscle and periosteum have specific physiological functions that can only be removed under specific disease conditions. (2) Due to technical limitations, we did not perform the 3D reconstruction of the fluorescence signal. However, 2D imaging applied in this research can quickly obtain the fluorescence images of the tumor tissue from various angles by adjusting the position of the tissue sample.

In conclusion, orbital adipose serves as the optimal control tissue for back-table fluorescence imaging of orbital tumors, with its size being the sole determinant of fluorescence imaging outcomes.

## Figures and Tables

**Figure 1 bioengineering-11-00922-f001:**
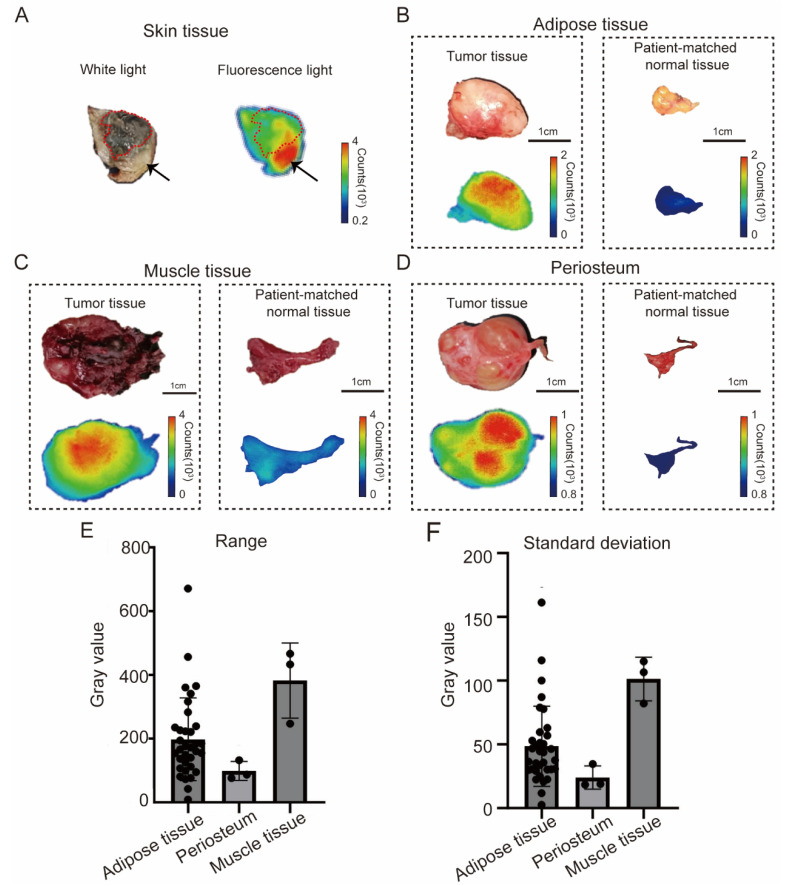
Fluorescence signal distribution in different control tissues. (**A**) White light and fluorescence imaging results in a representative skin tissue. The red dotted line outlines the diseased tissue and the arrow indicates the normal tissue. (**B**–**D**) White light and fluorescence imaging results in tumor tissues and patient-matched normal tissues (adipose, muscle, and periosteum tissues). (**E**,**F**) The range and standard deviation of gray values in the fluorescence imaging results of adipose, periosteum, and muscle tissues.

**Figure 2 bioengineering-11-00922-f002:**
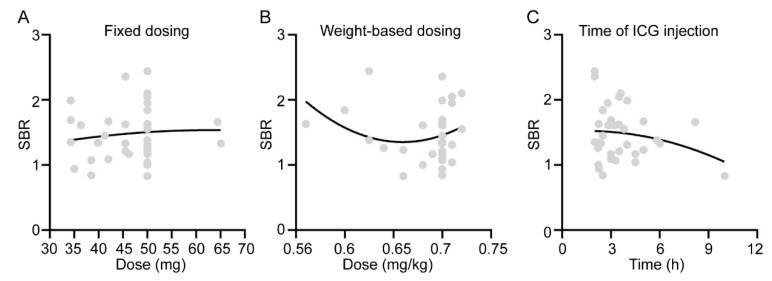
Correlation between the SBR of adipose tissue and the dose of ICG (**A**,**B**) or the time of ICG injection (**C**).

**Figure 3 bioengineering-11-00922-f003:**
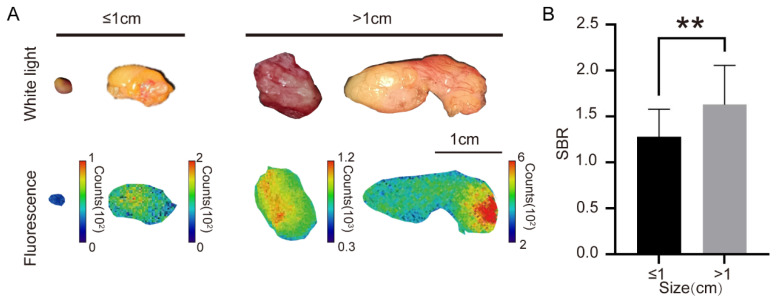
The fluorescence imaging results of adipose tissues might be affected by their sizes. (**A**) In four typical adipose tissues, the large ones (>1 cm) showed stronger fluorescence than the small ones (≤1 cm). (**B**) The calculated SBR of adipose tissues was also in proportion to the tissue size. The SBR of large tissues (>1 cm) is significantly higher than that of small tissues (≤1 cm). ** *p* < 0.01.

**Table 1 bioengineering-11-00922-t001:** Comparisons of clinical variables for SBRs (*n* = 34).

Variable		Mean SBR	*p*
Age	≤40 years (26%)	1.30	N.S.
	>40 years (74%)	1.52	
Gender	Female (50%)	1.41	N.S.
	Male (50%)	1.52	
Weight	≤65 kg (44%)	1.41	N.S.
	>65 kg (56%)	1.50	
Course	≤2 years (68%)	1.33	N.S.
	>2 years (32%)	1.34	
Type of Disease	Originating from the lacrimal gland (21%)	1.33	N.S.
	Originating from the nerve (23%)	1.47	
	Originating from the lymphatic system (23%)	1.62	
	Others (33%)	1.46	

SBR: signal-to-background ratio. N.S.: not significant.

## Data Availability

Data are fully available upon reasonable request to the corresponding author.

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
