# Peer review of "Orbital Adipose Tissue: The Optimal Control for Back-Table Fluorescence Imaging of Orbital Tumors"

_bioengineering, 2024, doi:10.3390/bioengineering11090922_

Round 1
Reviewer 1 Report
Comments and Suggestions for Authors
This manuscript reports the identification of orbital adipose tissue as the optimal control for fluorescence imaging of orbital tumors among several candidate tissues from patients. The parameters for enhanced SBR were further investigated and large adipose tissue was the best choice. Clinically, the conclusion might be of value for diagnosis, but the mechanism or reason behind the finding was not clear.
Specific comments:
1. For the tissue imaging, did the authors remove blood from the tissues? Blood will have great impact for different tissues especially for fluorescence imaging.
2. The data points for adipose tissue were much more than periosteum and muscle tissues. I suggest the authors to test more sample for periosteum and muscle tissues. The variation for adipose tissue was quite big, please explain the possible reasons for this phenomenon.
3. More experimental details are needed. How to use ICG to stain the tissues? What’s the concentration of ICG? How long for staining? How to extract fluorescence signals from the tissues? How to calculate the SBR values? And so on.
4. To identify the tumors, did the authors use the average fluorescence or total fluorescence to get rid of the background signals? What’s the reason to choose this method? Please explain.
5. To achieve high SBR or improved fluorescence imaging, the authors need to discuss the possible factors for fluorescence imaging in Discussion section. To further compare the impact of different tissues, the authors can consider the results from the close publication for further discussion [10.1021/acsnano.9b06504].
Author Response
This manuscript reports the identification of orbital adipose tissue as the optimal control for fluorescence imaging of orbital tumors among several candidate tissues from patients. The parameters for enhanced SBR were further investigated and large adipose tissue was the best choice. Clinically, the conclusion might be of value for diagnosis, but the mechanism or reason behind the finding was not clear.
Q1: For the tissue imaging, did the authors remove blood from the tissues? Blood will have great impact for different tissues especially for fluorescence imaging.
[REPLY]: Many thanks for the comments. Prior to conducting fluorescence imaging, we removed surface blood contamination from the tissues but did not specifically extract blood from within the tissues. Considering the potential impact of blood on fluorescence imaging, we performed a pathological examination of adipose tissue, periosteum tissue, and muscle tissue to compare the vascular density within these tissues. Hematoxylin and eosin (HE) staining revealed no significant differences in the number of blood vessels among the three tissue types (Figure R1). Consequently, we infer that the influence of blood vessels on the fluorescence imaging of these 3 tissues is minimal.
Figure R1. HE examination of adipose tissue (A), periosteum tissue (B) and muscle tissue (C). Capillaries are indicated by the asterisks.
Q2: The data points for adipose tissue were much more than periosteum and muscle tissues. I suggest the authors to test more sample for periosteum and muscle tissues. The variation for adipose tissue was quite big, please explain the possible reasons for this phenomenon.
[REPLY]: Many thanks for the comment. In orbital tumor surgery, adipose tissue is abundant and can be excised in small quantities without typically causing harm to patients. Conversely, the periosteum and muscle possess specific physiological functions; thus, their removal is not warranted unless there is a suspicion of tumor invasion. Consequently, due to ethical considerations and the imperative to protect patient well-being, obtaining larger quantities of periosteum and muscle tissue is challenging. We have elucidated this limitation in the Discussion section of revised manuscript.
Limitations: 1) The quantity of muscle and periosteum is considerably lower compared to adipose tissue, because muscle and periosteum have specific physiological functions that can only be removed under specific disease conditions.
For the variation for adipose tissue, we reviewed the fluorescence imaging results of all adipose tissues and found that the adipose tissue from patient 2#, 23# and 30# had greater range and standard deviation of gray values. Further pathological examination of these tissues revealed that they contained a higher abundance of neutrophils compared to other tissues (Figure R2). We speculated that the variation was due to differences in the content of inflammatory cells.

Figure R2. HE examination of three adipose tissues from 2# (A), 23# (B) and 30# patient (C). Blood vessels are indicated by the asterisks. Currently, neutrophils are distributed in the vascular wall.
Q3: More experimental details are needed. How to use ICG to stain the tissues? What’s the concentration of ICG? How long for staining? How to extract fluorescence signals from the tissues? How to calculate the SBR values? And so on.
[REPLY]: Many thanks for the comment. We have added related experimental details in the revised manuscript.
Prior to surgery, patients were administered an intravenous infusion of about 0.7 mg kg-1 body weight of ICG (Dandong Yichuang Pharmaceutical, China) 2 hours before undergoing resection of the orbital tumor.
Signal-to-background ratio (SBR) was quantitatively calculated by comparing the MFI of the excised tissue with that of black paper placed underneath the tissue, which served as the background.
Q4: To identify the tumors, did the authors use the average fluorescence or total fluorescence to get rid of the background signals? What’s the reason to choose this method? Please explain.
[REPLY]: Many thanks for the comment. To identify the tumors, we used the average fluorescence to get rid of the background signals. This is because when calculating SBR, different ROI areas are often selected based on the specific tissue areas. Utilizing average fluorescence for the calculation of SBR can mitigate the influence of differing ROI sizes. In contrast, employing total fluorescence for SBR calculation is typically susceptible to variations in ROI dimensions, thereby introducing potential biases.
Q5: To achieve high SBR or improved fluorescence imaging, the authors need to discuss the possible factors for fluorescence imaging in Discussion section. To further compare the impact of different tissues, the authors can consider the results from the close publication for further discussion [10.1021/acsnano.9b06504].
[REPLY]: Thank you for the comment. The sentence in the discussion may be clearer: Over the past decade, researchers have continuously improved fluorescent molecular imaging techniques, including the design of high-performance imaging molecules24 and other non-invasive imaging technologies.25,26

Reviewer 2 Report
Comments and Suggestions for Authors
The paper evaluates the back-table fluorescence imaging performance of different normal tissues and identify the optimal normal tissue in a cohort of 39 patients with orbital tumors. The authors declare that the findings of the invetogation demonstrated that adipose tissue consistently exhibited homogeneous hypofluorescence during back-table fluorescence imaging, regardless of patient clinical variables or imaging parameters. And that the size of adipose tissue was identified as the primary factor influencing its fluorescence imaging characteristics, supporting its utility as an ideal
control tissue for back-table fluorescence imaging.
Such findings are not new, so that the paper seems not to be particularly original. In addition the authors do not use imaging tools to enhance the precision of fluorescent method that could help in surgery excision of sample improving the medicine of precision.
In turn, their samples are 3D volumes, what about the 3D reconstruction of fluorescence signal in order to have SBR of the whole sample volume? What happen inside the samples?
I strongly suggest the authors to improve their paper by focusing on these key questions.
Author Response
Reply to comments of Reviewer #2:
The paper evaluates the back-table fluorescence imaging performance of different normal tissues and identify the optimal normal tissue in a cohort of 39 patients with orbital tumors. The authors declare that the findings of the invetogation demonstrated that adipose tissue consistently exhibited homogeneous hypofluorescence during back-table fluorescence imaging, regardless of patient clinical variables or imaging parameters. And that the size of adipose tissue was identified as the primary factor influencing its fluorescence imaging characteristics, supporting its utility as an ideal control tissue for back-table fluorescence imaging.
Specific comments:
Q1: Such findings are not new, so that the paper seems not to be particularly original. In addition the authors do not use imaging tools to enhance the precision of fluorescent method that could help in surgery excision of sample improving the medicine of precision.
[REPLY]: Many thanks for the comments. Precise resection of orbital tumors is a critical clinical necessity. Back table fluorescence imaging represents a real-time intraoperative technique for tumor identification, with the potential to facilitate accurate resection of orbital tumors[1]. The selection of appropriate control tissues is crucial for improving the accuracy of back table fluorescence imaging in tumor identification. However, the optimal control tissue for this imaging technique remains undetermined. To address this issue, we conducted a comparative analysis of the fluorescence imaging characteristics of commonly used control tissues, including periosteum, muscle, and adipose, for the first time. Additionally, we examined the fluorescence imaging properties of adipose tissue under various conditions, leading us to propose that adipose tissue serves as an ideal control tissue. While previous studies have indicated the potential of adipose tissue as a control tissue for back table fluorescence imaging, this is the inaugural study to systematically compare the fluorescence imaging characteristics across multiple commonly used control tissues. Furthermore, given that our research centers on the fluorescence imaging characteristics of various control tissues, we have employed widely accepted imaging and fluorescence analysis methodologies to enhance the representativeness of this study. We greatly appreciate the reviewers' comment and will prioritize the development of novel imaging tools in our future research endeavors.
References:
[1] Hu Z, Fang C, Li B, et al. First-in-human liver-tumour surgery guided by multispectral fluorescence imaging in the visible and near-infrared-I/II windows. Nature Biomedical Engineering. 2019;4(3):259-271.
Q2: In turn, their samples are 3D volumes, what about the 3D reconstruction of fluorescence signal in order to have SBR of the whole sample volume? What happen inside the samples? I strongly suggest the authors to improve their paper by focusing on these key questions.
[REPLY]: Thanks for the valuable suggestions. We agree that the ability to perform 3D imaging would certainly lead to better clinical detection outcomes.[1] However, at present, achieving 3D detection from optical imaging still poses significant technical challenges, with long processing times and poor precision.[2] Even when 3D results are obtained, the accuracy of the imaging may not be sufficient to support high-precision clinical decision-making.
Although 2D imaging applied in this paper cannot directly visualize the entire interior of the tissue, it is possible to quickly obtain fluorescence images of the tumor tissue from various angles by adjusting the position of the tissue sample. When these images from different angles are fused together, they can also efficiently and effectively reflect the internal conditions of the tumor.[3]
We have clarified the limitation in the Discussion section of the revised manuscript (page 7, line 235-238).
2)Due to technical limitations, we did not perform 3D reconstruction of the fluorescence signal. However, 2D imaging applied in this research can quickly obtain fluorescence images of the tumor tissue from various angles by adjusting the position of the tissue sample.
References:
[1] Zhao Z, Zhou Y, Liu B, , et al. Two-photon synthetic aperture microscopy for minimally invasive fast 3D imaging of native subcellular behaviors in deep tissue. Cell. 2023 May 25;186(11):2475-2491.e22.
[2] van Ineveld RL, Collot R, Román MB, et al. Multispectral confocal 3D imaging of intact healthy and tumor tissue using mLSR-3D. Nat Protoc. 2022 Dec;17(12):3028-3055.
[3] van Keulen S, Nishio N, Birkeland A, et al. The Sentinel Margin: Intraoperative Ex Vivo Specimen Mapping Using Relative Fluorescence Intensity. Clinical Cancer Research. 2019;25(15):4656-4662.

Reviewer 3 Report
Comments and Suggestions for Authors
This is a nice technical paper, which needs to be put in the broader perspective. Advantages and also potential shortcomings of FA-ICG need to be discuss. Please relate Introduction and Discussion to the following statement: "Among the advanced visualization systems, fluorescence angiography utilizing indocyanine green (FA-ICG) has emerged as an objective tool for evaluating intraoperative perfusion. Despite its versatility, FA-ICG imaging has limitations: for example, it requires external dye injection, is constrained by pharmacokinetic factors in repeat assessments, and may potentially lead to allergic reactions to the dye. To overcome these shortcomings, novel imaging techniques have been explored for microvascular imaging."
It would make sense to look at the references contained, for example in 10.2478/raon-2022-0051 where FA-ICG and HSI were complementary used, or 10.1007/s11605-023-05855-x where FA-ICG and LSCI are employed. My advice that even if papers are technical it is always important to paint wider picture and always to answer the question: "How relevant is my paper for a clinical setting?"
Comments on the Quality of English LanguageEnglish should be improved.
Author Response
Reply to comments of Reviewer #3:
Q1:This is a nice technical paper, which needs to be put in the broader perspective. Advantages and also potential shortcomings of FA-ICG need to be discuss. Please relate Introduction and Discussion to the following statement: "Among the advanced visualization systems, fluorescence angiography utilizing indocyanine green (FA-ICG) has emerged as an objective tool for evaluating intraoperative perfusion. Despite its versatility, FA-ICG imaging has limitations: for example, it requires external dye injection, is constrained by pharmacokinetic factors in repeat assessments, and may potentially lead to allergic reactions to the dye. To overcome these shortcomings, novel imaging techniques have been explored for microvascular imaging."
[REPLY]: Thank you very much for your suggestion. We introduced some novel imaging techniques in the discussion (page 6, line 190-193), which has been highlighted in red and shown below.
“Over the past decade, researchers have continuously improved fluorescent molecular imaging techniques, including the design of high-performance imaging molecules24 and other non-invasive imaging technologies.25,26”
Q2: My advice that even if papers are technical it is always important to paint wider picture and always to answer the question: "How relevant is my paper for a clinical setting?"
[REPLY]: Thank you very much for your suggestion. Precise resection of orbital tumors is a critical clinical necessity. Back table fluorescence imaging represents a real-time intraoperative technique for tumor identification, with the potential to facilitate accurate resection of orbital tumors. The selection of appropriate control tissues is crucial for improving the accuracy of back table fluorescence imaging in tumor identification. However, the optimal control tissue for this imaging technique remains undetermined. To address this issue, we conducted a comparative analysis of the fluorescence imaging characteristics of commonly used control tissues, including periosteum, muscle, and adipose, for the first time. Additionally, we examined the fluorescence imaging properties of adipose tissue under various conditions, leading us to propose that adipose tissue serves as an ideal control tissue. This study demonstrates significant potential in enhancing the accuracy of tumor identification during back table fluorescence imaging, thereby facilitating its clinical translation.
We also painted wider picture at the end of discussion (page 7, line 233-236) as your suggestion.
“This study contributes to enhancing the precision of NIR-II back-table imaging for tumor identification and facilitates its clinical application.”

Round 2
Reviewer 1 Report
Comments and Suggestions for Authors
This work has been well revised and is ready for publication.
Reviewer 2 Report
Comments and Suggestions for Authors
The authors addressed all the issues evidenced after first revision.